# Lack of Skeletal Effects in Mice with Targeted Disruptionof Prolyl Hydroxylase Domain 1 (*Phd1*) Gene Expressed in Chondrocytes

**DOI:** 10.3390/life13010106

**Published:** 2022-12-30

**Authors:** Weirong Xing, Destiney Larkin, Sheila Pourteymoor, William Tambunan, Gustavo A. Gomez, Elaine K. Liu, Subburaman Mohan

**Affiliations:** 1Musculoskeletal Disease Center, Loma Linda VA Healthcare System, Loma Linda, CA 92357, USA; 2Department of Medicine, Loma Linda University, Loma Linda, CA 92354, USA

**Keywords:** prolyl hydroxylase domain-containing protein, PHD protein, *Phd1* gene, osteoblast, chondrocyte, knockout mice, phenotype, hypoxia-inducible factor, endochondral bone formation

## Abstract

The critical importance of hypoxia-inducible factor (HIF)s in the regulation of endochondral bone formation is now well established. HIF protein levels are closely regulated by the prolyl hydroxylase domain-containing protein (PHD) mediated ubiquitin-proteasomal degradation pathway. Of the three PHD family members expressed in bone, we previously showed that mice with conditional disruption of the *Phd2* gene in chondrocytes led to a massive increase in the trabecular bone mass of the long bones. By contrast, loss of *Phd3* expression in chondrocytes had no skeletal effects. To investigate the role of *Phd1* expressed in chondrocytes on skeletal development, we conditionally disrupted the *Phd1* gene in chondrocytes by crossing *Phd1 floxed* mice with *Collagen 2α1-Cre* mice for evaluation of a skeletal phenotype. At 12 weeks of age, neither body weight nor body length was significantly different in the *Cre^+^*; *Phd1^flox/flox^* conditional knockout (cKO) mice compared to *Cre^−^*; *Phd1^flox/flox^* wild-type (WT) control mice. Micro-CT measurements revealed significant gender differences in the trabecular bone volume adjusted for tissue volume at the secondary spongiosa of the femur and the tibia for both genotypes, but no genotype differences were found for any of the trabecular bone measurements of either femur or tibia. Similarly, cortical bone parameters were not affected in the *Phd1* cKO mice compared to control mice. Histomorphometric analyses revealed no significant differences in bone area, bone formation rate or mineral apposition rate in the secondary spongiosa of femurs between cKO and WT control mice. Loss of *Phd1* expression in chondrocytes did not affect the expression of markers of chondrocytes (*collage 2*, *collagen 10*) or osteoblasts (*alkaline phosphatase*, *bone sialoprotein*) in the bones of cKO mice. Based on these and our published data, we conclude that of the three PHD family members, only *Phd2* expressed in chondrocytes regulates endochondral bone formation and development of peak bone mass in mice.

## 1. Introduction

Previous studies have uncovered that the prolyl hydroxylase domain-containing proteins (PHDs) are negative regulators of the hypoxia-inducible transcription factor (HIF)1α [1,2]. The hydroxylation of specific proline residues (Pro-402 and Pro-564) in the C-terminal oxygen-dependent degradation domains of the HIF1α by PHDs, primarily the PHD2 isoform, leads to the targeting of HIF1α for ubiquitination through an E3 ligase complex initiated by the binding of the Von Hippel Lindau protein (pVHL) and subsequent proteasomal degradation [1,2]. When the oxygen level is low in the cells, the *Phd* gene expression is suppressed, and the HIF1α degradation is reduced and the protein is accumulated in the cytoplasm from where it traffics to nucleus and binds to HIF regulatory elements in the promoter regions of the hypoxia-responsive genes including VEGF, Runx2 and osterix to regulate the target gene expression and subsequently bone formation [3,4]. In mammals, PHD enzymes include PHD1, PHD2, and PHD3 [5]. Both PHD1 and PHD2 contain more than 400 amino acid residues while PHD3 has less than 250. All three members contain the highly conserved hydroxylase domain in the catalytic carboxy-terminal region and are expressed in bones. However, PHD1 and PHD2 preferably hydroxylate the N-terminal oxygen-dependent degradation domains (NODD) but are less active for the C-terminal oxygen-dependent degradation domains (CODD) whereas PHD3 almost exclusively hydroxylates the CODD [6,7]. Mice with deletion of *Phd1* and *Phd3* genes grow normal, but *Phd2* gene knockout (KO) in mice causes embryonic lethality because the placenta is underdeveloped [8]. The structural difference among the PHD proteins and the data from mouse genetic studies suggest they may have tissue specific functions.

We previously unveiled that PHD2 was highly expressed in bone cells and contributed to an indispensable role in regulating bone homeostasis by upregulating the transcription of genes critical for osteoblast differentiation and function [9]. Mice with targeted deletion of *Phd2* in osteoblasts were smaller and died 12 to 14 weeks after birth. Bone mineral density (BMD) in femurs and the ratio of trabecular bone volume to the tissue volume (BV/TV) in the secondary spongiosa regions of the long bones of the osteoblast-specific conditional knockout (cKO) mice were dramatically low [9]. Mice lacking PHD2 protein in chondrocytes born normally, but the growth after birth were retarded because of elevated mineralization of the cartilage matrix. The chondrocyte-specific cKO mice manifested an increased endochondral bone formation in the femur, tibia and spine, resulting from increased HIF signaling in chondrocytes [10]. While the expression level of *Phd3* in the bones in chondrocyte specific *Phd2* KO mice was dramatically elevated, loss of *Phd3* in chondrocytes did not affect endochondral bone formation and skeletal phenotypes [11]. To investigate the role of *Phd1* expressed in chondrocytes on skeletal development, we conditionally disrupted the *Phd1* gene in chondrocytes by crossing *Phd1 floxed* mice with *Col2α1-Cre* mice for evaluation of skeletal phenotypes.

## 2. Materials and Methods

### 2.1. Breeding Strategy of cKO Mice

*Phd1 floxed* mice were bred with mice overexpressing *Cre* under the control of the Collagen 2α1 (*Col2α1)* promoter to produce *Cre* positive, *Phd1 floxed* heterozygous mice (*Phd1^flo^*^x/+^; *Col2α1-Cre^+^*) according to the breeding strategy described previously [11,12,13]. The *Phd1^flox/+^*; *Col2α1-Cre^+^* mice were then backcrossed with *Phd1^flox/flox^* mice to generate *Cre*-positive, *loxP*-homozygous (*Phd1^flox/flox^*; *Col2α1-Cre+*) cKO and *Cre*-negative, *Phd1 loxP*-homozygous or heterozygous (*Phd1^flox/flox^*; *Phd1^flox/+^*) wild-type (WT) littermates (Figure 1A). The genetic background of these mice is C57BL/6. Both sex mice were used in this study. Mice were housed at the Loma Linda VA Healthcare System (Loma Linda, CA, USA) at 22 °C and with 14 h light and 10 h dark, as well as free access to food and water. Experiments were carried out according to the protocol approved by the Institutional Animal Care and Use Committee (IACUC) of the Loma Linda VA Healthcare System (CA, USA). Mice were anesthetized using isoflurane before tail clipping. Mice were euthanized by exposing to CO_2_ gas proceeded by cervical dislocation.

### 2.2. Evaluation of Bone Phenotypes

Areal BMD of the total body, long and lumbar bones (L4–6) of 12-week-old mice were quantified by the FAXITRON UltraFocus^DXA^ 1000 as reported [11,14,15]. Trabecular and cortical bones of the femur and the tibia were scanned and quantified by microcomputed tomography (µCT) in 12-week old mice described previously [16]. The formalin-fixed bones in PBS were scanned by µCT with 55 kVp volts and a voxel size of 10.5 µm. A 1.05 mm cortical bone in the mid-diaphysis of the femur and the tibia were analyzed for cortical bone parameters. A 2.1 mm of the secondary spongiosa of the distal femur and the proximal tibia beginning 0.3675 mm from the growth plates were assessed for TV(mm^3^), BV(mm^3^), and BV/TV, as described [17,18,19].

### 2.3. Double Labeling and Histomorphometric Analyses

Twelve-week-old mice were injected intraperitoneally with calcein (20 mg/kg) eight and two days before euthanization by CO_2_ to label mineralizing bone surfaces. Mouse right femurs were fixed in 10% formalin for 3 days, washed 3 times with PBS, dehydrated, and embedded in methyl methacrylate resin for sectioning. The sampling sites and histomorphometric analyses were performed as described [19]. The first and the second calcein labeling of the trabecular bone in the secondary spongiosa region of the distal femurs were blindly quantified with OsteoMeasure V3.1.0.2 computer software (OsteoMetrics, Decatur, GA, USA) [20,21]. The mineral apposition rate (MAR) and bone formation rate/bone surface (BFR/BS) were calculated as described previously [22].

### 2.4. Primary Chondrocyte Culture

Primary chondrocytes isolated from the rib cartilage and the growth plates of the femurs and the tibias of 10-day old WT and cKO mice (3 female and 3 male littermate mice) were cultured as previously described [23]. Cells were grown in DMEM/F12 medium containing 10% fetal bovine serum (FBS), penicillin (100 U/mL), and streptomycin (100 μg/mL) to approximately 90% confluence before harvesting for RNA extraction.

### 2.5. RNA Extraction and Real-Time PCR

Total RNA was extracted from the femurs and the tibias of the WT and cKO mice or primary chondrocyte cultures derived from WT and cKO mice with the Trizol as described [24,25]. An aliquot of RNA (300 ng) was reverse-transcribed into cDNA in 20 µL volume of reaction by oligo(dT)_12–18_ primer. A real-time PCR contained 0.5 µL template cDNA, 1x SYBR GREEN master mix (Qiagen), and 100 nM of specific forward and reverse primers in a 25 μL volume of reaction. Primers used for real-time PCR are listed in Table 1. Relative gene expression was calculated by using the ^ΔΔ^CT method [26].

### 2.6. Statistical Analysis

Student’s *t*-test was used for data analyses. Data are Mean ± SEM (*n* = 6–10).

## 3. Results

### 3.1. Expression of Phd1 Was Partially Disrupted in Chondrocytes in cKO Mice

To test if loss of *Phd1* expression in chondrocytes impairs endochondral bone formation, we produced chondrocyte-specific *Phd1* cKO mice by breeding the *Phd1 floxed* mice with the *Col2α-Cre* mice, in which the Cre recombinase is overexpressed in *Col2α*-expressing chondrocytes [13,27]. After 2 generations of breeding, the *Phd1 floxed, Cre^+^* cKO mice (*Phd1^flox/flox^*; *Col2α-Cre^+^*) were produced and compared to *Cre* negative WT littermates (*Phd1^flox/flox^* or *Phd1^lox/+^*; *Cre*^−^). The cKO mice born and developed normally. To investigate if PHD1 protein exists in bone cells of cKO mice, RNA was isolated from chondrocytes derived from the growth plates of the femurs and the tibias, and the ribs of 10-day old WT and cKO mice for real-time PCR with specific primers to *Phd1, 2* and *3*. As shown in Figure 1B, the expression levels of *Phd1* transcript were reduced by 66% and 45% in the growth plate and rib chondrocytes, respectively, in the cKO mice compared to WT mice. By comparison, *Phd2* was increased in chondrocytes of both growth plates and ribs of *Phd1* cKO mice by 79% and 41%, respectively. While the expression levels of Phd3 were 56% and 33% higher, respectively, in the growth plate and rib chondrocytes of *Phd1* cKO mice, only *Phd3* expression level in growth plate chondrocytes was significantly higher compared to WT mice.

### 3.2. Deletion of Phd1 in Col2α-Expressing Chondrocytes Does Not Affect Skeletal Growth in Mice at 12 Weeks of Age

To analyze the bone phenotypes, we performed DXA screening and μCT scanning. At 12 weeks after birth, neither body weight nor body length was significantly different in the cKO mice compared to gender-matched control mice for either gender (Figure 2A). DXA measurements revealed no significant changes in total body, femur, tibia, and lumbar BMDs between the two genotypes for either gender (Figure 2B,C). Concurred with DXA data, µCT scanning of the femoral trabecular bone uncovered no significant changes in either BV/TV or any of the trabecular bone parameters including BMD, trabecular number (Tb. N), trabecular thickness (Tb. Th) and trabecular spacing (Tb. Sp) in *Phd1* cKO from the gender-matched WT mice for either gender (Figure 3A–C). The tibial BMD, BV/TV, Tb. N, Tb. Th, and Tb. Sp in cKO mice were also comparable to WT mice in either gender mice (Figure 4A–C). Deletion of the *Phd1* gene in chondrocytes had no impact on either cortical BV/TV ratio or BMD (Figure 5A). The deficiency of PHD1 expression had no effect on cortical BV/TV and BMD in the tibia either (Figure 5B).

### 3.3. Knockout of Phd1 in Chondrocytes Neither Influences Bone Formation Nor Expression of Marker Genes of Osteoblast/Chondrocyte Differentiation

To determine if the deletion of *Phd1* in *Col2α1* expressing cells impacts osteoblast formation, and trabecular bone formation in the femur, we performed histomorphometry analyses and examined the bone marker genes expression in long bones of the cKO mice. We uncovered that knockdown of *Phd1* expression in chondrocytes neither affected the MAR) nor the BFR/BS in cKO mice as compared to the WT control littermates (Figure 6A). Consistent with the histomorphometric data, lack of *Phd1* in chondrocytes had no impact on the differentiation of both osteoblasts and chondrocytes as evidenced by comparable expression levels of marker genes, *alkaline phosphatase (Alp), bone sialoprotein (Bsp), collagen 2 (Col2)*, and *collagen 10 (Col10)* in long bones in the cKO mice compared with the WT mice (Figure 6B).

## 4. Discussion

Of three family members, the PHD2 protein is the most abundant in bones [28]. PHD2 is believed to be the critical oxygen sensor during hypoxia, which is emphasized by the fact that mice with global deletion of the *Phd2* gene are embryonically lethal [8]. By contrast, mice with global disruption of either the *Phd1* or *Phd3* gene develop normally. Consistent with an important role for PHD2 in bones, we and others have shown that disruption of the *Phd2* gene in osteoblasts and chondrocytes influenced bone formation and development of peak bone mass [9,27,29]. Mice with deletion of *Phd2* in osteoblasts were smaller and died twelve to fourteen weeks after birth. Femoral BMD and trabecular BV/TV of osteoblast-specific cKO mice were notably diminished. By contrast, mice lacking *Phd2* were born normally, but the development was retarded after birth resulted from abnormal mineralization of the cartilage matrix. Endochondral bone formation was enhanced in the femur, tibia, and spine of the *Phd2* chondrocyte-specific cKO mice [10]. While the expression level of *Phd3* elevated 7-fold in chondrocytes of *Phd2-c*KO mice, targeted disruption of *Phd3* gene in mice had no impact on bone cell differentiation, endochondral bone formation, and bone development [11]. Our previous studies indicate that *Phd3*, unlike *Phd2*, does not play an important role in regulating chondrocyte differentiation and bone growth.

PHD enzymes function through hydroxylation of the specific proline and asparagine residues of HIF-α and negatively regulate HIF-α protein stability [30]. Both HIF1α and HIF2α contain two prolyl hydroxylation sites in a central degradation domain of HIF1α. Hydroxylation of these sites promotes HIF1α interaction with the ubiquitin ligase for ubiquitination and subsequent degradation [5,31]. Hydroxylation of an asparagine residue in the C-terminals prevents HIFα transcription factor from cooperation with the co-activator, p300/CBP, leading to HIF1α inactivation [32]. Recent studies suggest that the PHD1/2 proteins specifically and preferentially hydroxylate their substrates. Although both PHD1 and PHD2 are active on CODD and NODD, PHD1 appears to act more effectively on substrate HIF2α, whereas PHD2 more actively hydroxylates on substrate HIF1α [7]. These studies indicate the PHD1 vs. PHD2 hydroxylate HIFα in a CODD sequence-dependent manner. Congruent with these studies, *Phd2* deletion mice had an increased level of HIF1α in the liver and the kidneys but no increase in the HIF2α protein was noted. In contrast, PHD1/3 double deficient mice had elevated level of HIF2α protein only in the liver [28,33]. On the other hand, loss of *Hif1α* in osteoblasts impaired skeletal growth [34,35]. Mice without HIF1α protein in the condensing mesenchymal stem cells had shorter bones, impaired mineralized skulls and wider sutures because of severe chondrocyte apoptosis and impaired chondrocyte proliferation in the growth plate [34]. By contrast, loss of *HIF2α* protein in mice only resulted in a modest decrease in trabecular BV [36]. However, recent mouse genetics studies demonstrated that HIF2 is a negative regulator of osteoblastogenesis and bone mass accrual by upregulating the transcription factor SOX9 to impair osteoblast differentiation [37]. Loss of HIF2 in mesenchymal progenitors increases bone mass by promoting bone formation without affecting bone resorption [37,38]. Since SOX9 is also a master transcription factor in chondrocyte differentiation, we assumed that if the PHD1/HIF/SOX9 signaling axis in chondrocytes is important in endochondral bone formation, then loss of the *Phd1* gene in chondrocytes should influence trabecular bone mass because PHD1 hydroxylates target HIF proteins and promotes ubiquitin-mediated protein degradation. To test the hypothesis, *Phd1* was deleted in chondrocytes by breeding *Phd1 floxed* mice with *Col2α1-Cre* mice, and the effects of knocking out the *Phd1* gene in chondrocytes on the development of peak bone mass was evaluated. Surprisingly, no significant changes in either body weight or body length was observed in the cKO mice compared to gender- and age-matched WT littermates. Micro-CT measurements unveiled significant gender differences in the trabecular BV/TV at the metaphysis of either the femur or the tibia of WT and cKO mice. We did not observe a genotype difference for any of the trabecular measurements of the long bones. Similarly, cortical bone parameters were not affected in the *Phd1* cKO mice compared to control mice. Histomorphometric analyses observed no significant differences in bone formation rate or mineral apposition rate in the secondary spongiosa of femurs between cKO and WT control mice. These data suggest that *Phd1* expressed in chondrocytes exert no major role in regulating the skeletal phenotype.

We found that *Phd1* expression was reduced only by 66% and 45%, respectively, in cultured growth plate and rib chondrocytes derived from the long bones of 10-day old cKO mice. The magnitude of reduction in *Phd1* expression in growth plate chondrocytes of *Phd1* cKO mice was similar to the 60% reduction in *Phd2* expression reported previously in the growth plate chondrocytes of *Phd2* cKO mice [27]. One potential explanation for the partial reduction in *Phd1* expression in the cKO mice is the possibility that the cultures used were not entirely homogeneous for chondrocytes and might contain other cell types (fibroblasts, osteoblasts) which remains to be examined. In any case, our data show that 66%-45% loss of *Phd1* transcript in the growth plate and rib chondrocytes had no impact on the transcription of chondrocyte markers, *Col2* and *Col10*, or osteoblast markers, *Alp*, *Bsp2*, in the bones of cKO mice. By contrast, we found that the expression levels of *Phd2* and *Phd3* were increased in the chondrocytes derived from *Phd1* cKO mice which could represent a compensatory response to the loss of *Phd1* expression. In previous studies, we reported that PHD2 was a negative regulator of chondrocyte differentiation since disruption of *Phd2* gene in chondrocytes, promoted chondrocyte differentiation and increased trabecular bone formation [27,39]. We, therefore, anticipated an increased *Phd2* expression to reduce chondrocyte differentiation, and trabecular bone volume in the *Phd1* cKO mice. However, that was not the case. Further studies comparing the skeletal phenotypes *Phd1, Phd2* and *Phd1/2 cKO* mice are needed to verify if the compensatory increase in *Phd2* expression has any role in the *Phd1* cKO mice. While expression of *Phd3* was elevated by 56% in the growth plate chondrocytes, this compensatory increase in the expression of *Phd3* is unlikely to play a significant role in regulating bone formation based on our previous findings on the lack of skeletal phenotype in chondrocyte specific *Phd3* cKO mice. Consistent with our interpretation, Wu et al. found that the trabecular bone phenotype was unaffected in mice with disruption of both *phd1* and *Phd3* genes in osterix expressing cells [40]. Our data, together with our previous reports, imply that *Phd2* transcribed in chondrocytes is a major contributor to endochondral bone formation [27]. PHD2 expressed in chondrocytes can functionally compensate for the loss of PHD1 in *Phd1* cKO mice.

## 5. Conclusions

*Phd1* expressed in chondrocytes does not regulate endochondral bone formation. Of the *Phd1/2/3 genes*, only *Phd2* transcribed in chondrocytes contributes to the endochondral bone formation and the peak bone mass in mice.

## Figures and Tables

**Figure 1 life-13-00106-f001:**
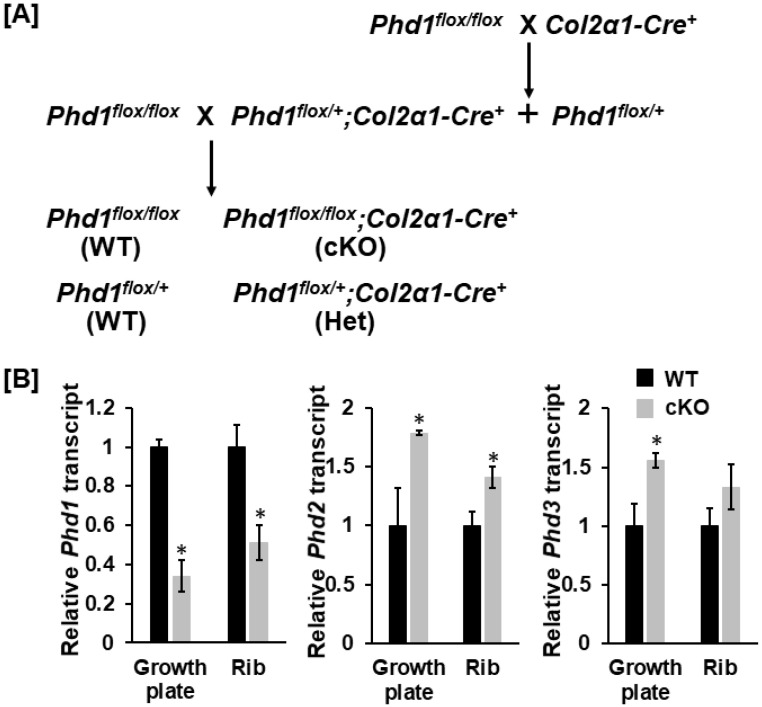
Breeding strategy of *Phd1* conditional knockout (cKO) mice. (**A**) A breeding strategy of *Phd1* cKO mice, heterozygous (Het) and wild-type (WT) mice. (**B**) *Phd1* expression was partially disrupted in chondrocyte cultures derived from the cKO mice. Total RNA was extracted from primary chondrocytes derived from the femur growth plates and the ribs of 10-day old mice for quantitative PCR (*n* = 3). Star (*): *p* < 0.01.

**Figure 2 life-13-00106-f002:**
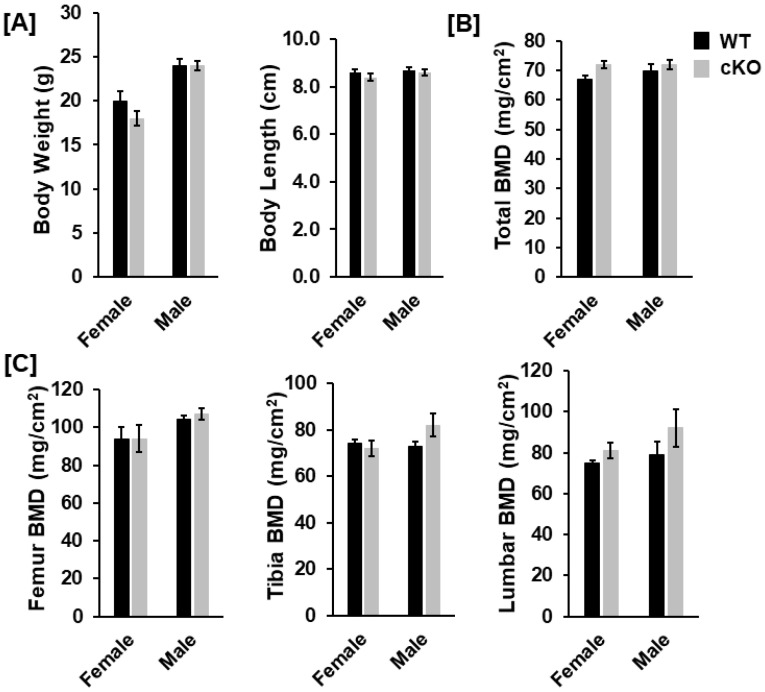
Bone parameters were not significantly altered in the *Phd1* cKO mice at 12 weeks after birth. (**A**) Body weight and length of the mice, as indicated in the figure. (**B**,**C**) Bone mineral density (BMD) for total body, the femur, the tibia, and the lumbar bone, respectively, quantified by DXA (*n* = 6–10).

**Figure 3 life-13-00106-f003:**
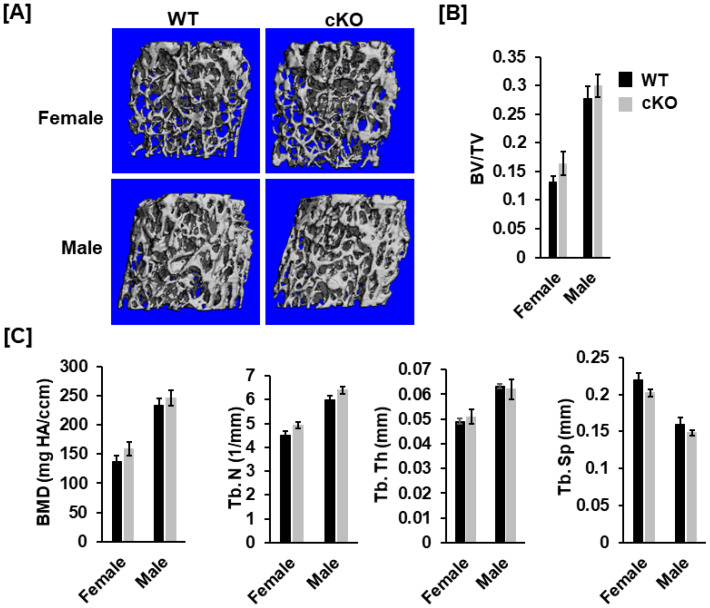
Trabecular parameters were not significantly altered in the femur of the *Phd1* cKO mice at 12 weeks after birth. (**A**) Representative µCT images of the distal femurs. (**B**,**C**) Trabecular bone data of the femurs (BV/TV, BMD, Tb.N, Tb. Th, and Tb. Sp) measured by µCT (*n* = 6–10). BV, bone volume; TV, tissue volume; Tb. N, trabecular number; Tb. Th, trabecular thickness; Th. Sp, Trabecular spacing.

**Figure 4 life-13-00106-f004:**
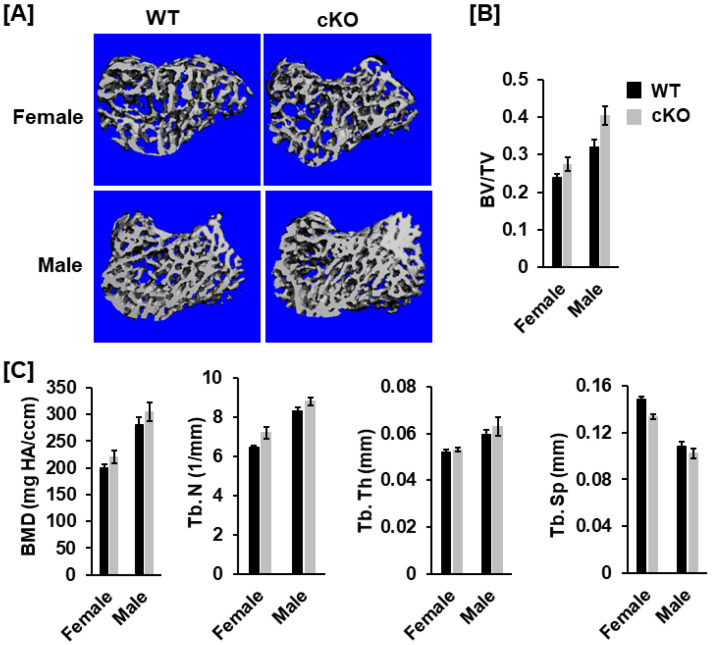
No changes in trabecular parameters were found in the proximal metaphysis of the tibia in the *Phd1* cKO mice at 12 weeks after birth. (**A**) Representative µCT images of the proximal tibias. (**B**,**C**) Trabecular bone data of the tibias (BV/TV, BMD, Tb.N, Tb. Th, and Tb. Sp) measured by µCT (*n* = 6–10).

**Figure 5 life-13-00106-f005:**
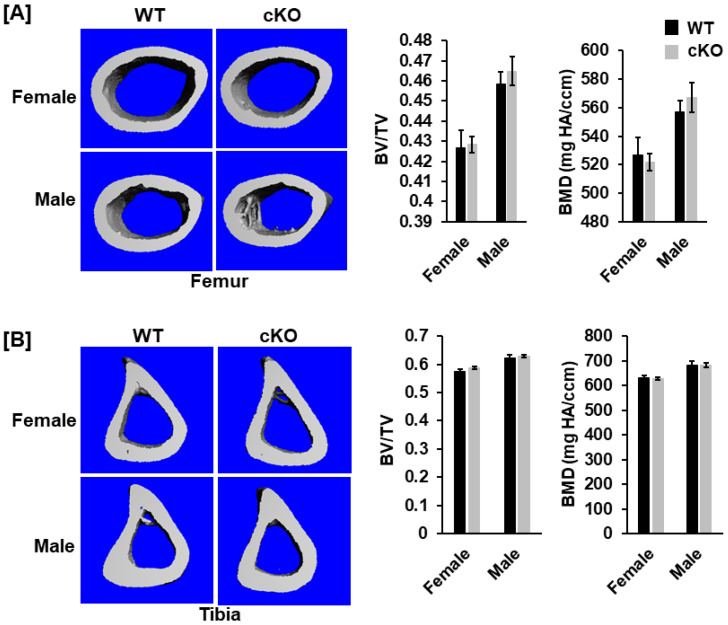
No changes were found in cortical bone parameters of the femur and the tibia in the *Phd1* cKO mice at 12 weeks after birth. (**A**) Representative µCT images of the cortical bone of the femurs and the quantitative data of the cortical bones of the femurs (BV/TV, BMD). (**B**) Images of the cortical bone of the tibias and the cortical bone data of the tibias (BV, BMD) (*n* = 6–10).

**Figure 6 life-13-00106-f006:**
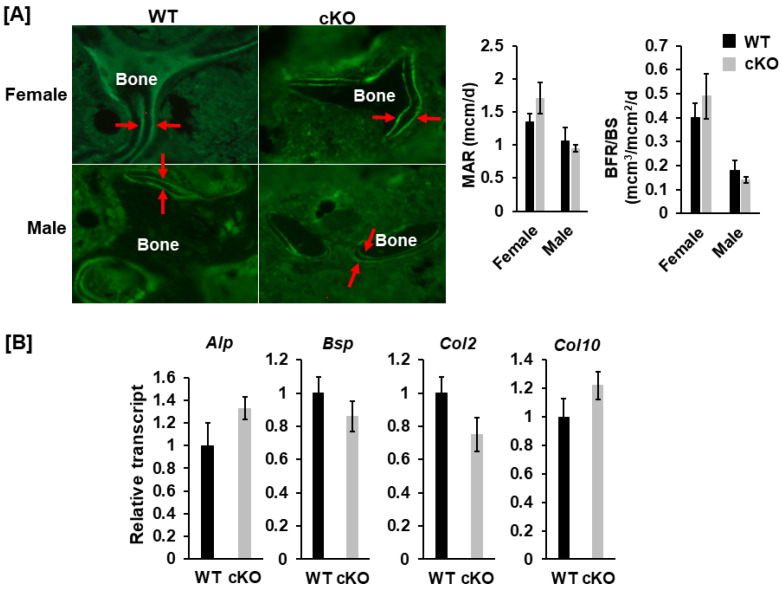
Knockout of *Phd1* in chondrocytes neither influences bone formation nor expression of markers of osteoblast/chondrocyte differentiation. (**A**) Representative images of the calcein-labeled trabecular bone of the distal femurs and the quantitative data of mineral apposition rate (MAR) and bone formation rate/bone surface (BFR/BS), respectively. The bone indicated by two red arrows is the double-labeled, newly formed bone (*n* = 6). (**B**) Expression levels of the marker genes of osteoblast and chondrocyte differentiation in long bones measured by RT-real-time PCR. *Alp*, *alkaline phosphatase*; *Bsp*, * bone sialoprotein*; *Col2*, *collagen 2*; *Col10*: *collagen 10*.

**Table 1 life-13-00106-t001:** Primer Sequence for Real-Time PCR.

Gene	Forward Primer	Reverse Primer
*Ppia*	5′-CCATGGCAAATGCTGGACCA	5′-TCCTGGACCCAAAACGCTCC
*Phd1*	5′-GGAACCCACATGAGGTGAAG	5′-AACACCTTTCTGTCCCGATG
*Phd3*	5′-GGGACGCCAAGTTACACGGA	5′-GGGCTCCACGTCTGCTACAA
*Phd2*	5′-GAAGCTGGGCAACTACAGGA	5′-CATGTCACGCATCTTCCATC
*Alp*	5′-ATGGTAACGGGCCTGGCTACA	5′-AGTTCTGCTCATGGACGCCGT
*Bsp*	5′-AACGGGTTTCAGCAGACAACC	5′-TAAGCTCGGTAAGTGTCGCCA
*Col2*	5′-TGGCTTCCACTTCAGCTATG	5′-AGGTAGGCGATGCTGTTCTT
*Col10*	5′-ACGGCACGCCTACGATGT	5′-CCATGATTGCACTCCCTGAA

Note: *Ppia, peptidylprolyl isomerase A; Phd, prolyl hydroxylase domain-containing protein; Alp, alkaline phosphatase; Bsp, bone sialoprotein; Col2, collagen 2; Col10, collagen 10*.

## Data Availability

Raw data are available upon request.

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
