# Peer review of "Lack of Skeletal Effects in Mice with Targeted Disruptionof Prolyl Hydroxylase Domain 1 (Phd1) Gene Expressed in Chondrocytes"

_life, 2022, doi:10.3390/life13010106_

Round 1

Reviewer 1 Report

The prolyl hydroxylase domain-containing protein (PHD) can regulate the ubiquitin-proteasomal degradation of hypoxia-inducible factors (HIFs), the important regulator of endochondral bone formation. The authors’ group has reported that among the three members of the PHD family, deficiency of PHD2 protein in chondrocytes results in growth retardation after birth because of elevated mineralization of the cartilage matrix, and in contrast, loss of expression of Phd3 in chondrocytes does not cause skeletal abnormality. In this manuscript, the authors generated mice with specific deletion of PHD1 in chondrocytes by crossing Phd1 floxed mice with Col2α1-Cre mice, and found that conditional disruption of the Phd1 gene in chondrocytes did not affect the growth of the mice, nor did the subsequent bone formation in the long bone. The manuscript added additional data to fully understand the role of the whole PHD protein family in bone development and diseases. However, the authors need to discuss further the effects of secondarily increased PHD2 expression in the bone caused by specific deletion of PHD1 in chondrocytes. In addition, the authors should carefully read through the whole manuscript to correct the typos, for example, “When the oxygen level is level…” in lines 46-47, and “Mice with targeted deletion of Phd2 in osteoblasts led were smaller and died 12 to 14 weeks after birth” in lines 64-65.

Author Response

  1. The authors need to discuss further the effects of secondarily increased PHD2 expression in the bone caused by specific deletion of PHD1 in chondrocytes.

Response: We thank the reviewer her/his comment. The expression level of Phd2 was increased by 79% and 41% in the growth plate and rib chondrocytes in Phd1 cKO mice.  In previous studies, we reported that PHD2 was a negative regulator of chondrocyte differentiation since disruption of Phd2 gene in chondrocytes, promoted chondrocyte differentiation and increased trabecular bone formation (PMID:26562260; PMID:27775044).  We, there, anticipated increased Phd2 expression to reduce chondrocyte differentiation, and trabecular bone volume in the Phd1 cKO mice.  However, that was not the case.  Further studies comparing the skeletal phenotypes Phd1, Phd2 and Phd1/2 cKO mice are needed to verify if the compensatory increase in Phd2 expression has any role in the Phd1 cKO mice.  While expression of Phd3 was elevated by 56% in the growth plate chondrocytes, this compensatory increase in the expression of Phd3 is unlikely to play a significant role in regulating bone formation based on our previous findings on the lack of skeletal phenotype in chondrocyte-specific Phd3 cKO mice. Consistent with our interpretation, Wu et al. (PMID: 25846796) found that the trabecular bone phenotype was unaffected in mice with disruption of both phd1 and Phd3 genes in osterix expressing cells.  We have added this information in the revised manuscript.

  1. The authors should carefully read through the whole manuscript to correct the typos, for example, “When the oxygen level is level…” in lines 46-47, and “Mice with targeted deletion of Phd2 in osteoblasts led were smaller and died 12 to 14 weeks after birth” in lines 64-65.

Response: We apologize for the typos. We have now corrected the errors in the revised manuscript.

Reviewer 2 Report

The manuscript by Xiong et al. showed the skeletal effect of disruption of Phd1 gene in chondrocytes using conditional knockout mouse model. 

Cre+Phd1flox/flox conditional knockout (cKO) mice have no significant skeletal phenotype compared with WT mice based on the microCT analysis and bone formation assay. However, Figure 1 showed that Phd1 is only reduced by 66% and 45% in the growth plate chondrocytes and rib chondrocytes, while Phd2 and Phd3 increased dramatically probably in a complementary way. This makes the situation complicated and it is not crystal clear if the phenotype is the real effect of targeting Phd1 gene.

Author Response

  1. Figure 1 showed that Phd1 is only reduced by 66% and 45% in the growth plate chondrocytes and rib chondrocytes, while Phd2 and Phd3 increased dramatically probably in a complementary way. This makes the situation complicated and it is not crystal clear if the phenotype is the real effect of targeting Phd1 gene.

Response: We thank the reviewer for the comment. Primary chondrocytes isolated from the rib cartilage and the growth plates of the femurs and the tibias of 10-day old WT and cKO mice were cultured and used for RNA extraction.  The magnitude of reduction in Phd1 expression in growth plate chondrocytes of Phd1 cKO mice was similar to the 60% reduction in Phd2 expression reported previously in the growth plate chondrocytes of Phd2 cKO mice (PMID: 26562260). One potential explanation for the partial reduction in Phd1 expression in the cKO mice is the possibility that the cultures used were not entirely homogeneous for chondrocytes and might contain other cell types (fibroblasts, osteoblasts) which remains to be examined.  In any case, we have revised the discussion and stated that “We found that 66%-45% loss of Phd1 transcript in the growth plate and rib chondrocytes had no impact on the transcription of chondrocyte markers, Col2 and Col10, or osteoblast markers, Alp, Bsp2, in the bones of cKO mice”.   Please see response to reviewer #1 regarding the potential complication of results caused by compensatory changes in Phd2 and Phd3 expression on the skeletal phenotype in the Phd1 cKO mice. 
